# Body Weight, Physical Activity, and Risk of Cancer in Lynch Syndrome

**DOI:** 10.3390/cancers13081849

**Published:** 2021-04-13

**Authors:** Tero Sievänen, Timo Törmäkangas, Eija K. Laakkonen, Jukka-Pekka Mecklin, Kirsi Pylvänäinen, Toni T. Seppälä, Päivi Peltomäki, Sarianna Sipilä, Elina Sillanpää

**Affiliations:** 1Gerontology Research Centre and Faculty of Sport and Health Sciences, University of Jyväskylä, P.O. Box 35 (VIV), 40014 Jyväskylä, Finland; timo.tormakangas@jyu.fi (T.T.); eija.k.laakkonen@jyu.fi (E.K.L.); sarianna.sipila@jyu.fi (S.S.); elina.sillanpaa@jyu.fi (E.S.); 2Department of Surgery, Central Finland Health Care District, 40620 Jyväskylä, Finland; jukka-pekka.mecklin@ksshp.fi; 3Faculty of Sport and Health Sciences, University of Jyväskylä, 40014 Jyväskylä, Finland; 4Department of Education, Central Finland Health Care District, 40620 Jyväskylä, Finland; kirsi.pylvanainen@ksshp.fi; 5Department of Surgical Oncology, Johns Hopkins University, Baltimore, MD 21218, USA; toni.t.seppala@hus.fi; 6Department of Surgery, Helsinki University Hospital, University of Helsinki, 00100 Helsinki, Finland; 7Department of Medical and Clinical Genetics, University of Helsinki, 00100 Helsinki, Finland; paivi.peltomaki@helsinki.fi; 8Institute for Molecular Medicine Finland, University of Helsinki, 00100 Helsinki, Finland

**Keywords:** epidemiology, hereditary non-polyposis colorectal cancer, lifestyle

## Abstract

**Simple Summary:**

Lifestyle modifies cancer risk in the general public. How lifestyle modifies cancer risk in individuals carrying the inherited pathogenic gene variants in DNA mismatch repair genes (Lynch syndrome) remains understudied. We conducted a retrospective study with cancer register data to investigate associations between body weight, physical activity, and cancer risk among Finnish Lynch syndrome carriers (*n* = 465, 54% women). The results of our study indicated that longitudinal weight gain increases cancer risk, whereas being highly physically active during adulthood could decrease cancer risk in men. Further, women were observed to be less prone to lifestyle-related risk factors than men. The results emphasize the role of weight maintenance and high-intensity physical activity throughout the lifespan, especially in men with Lynch syndrome.

**Abstract:**

Lynch syndrome (LS) increases cancer risk. There is considerable individual variation in LS cancer occurrence, which may be moderated by lifestyle factors, such as body weight and physical activity (PA). The potential associations of lifestyle and cancer risk in LS are understudied. We conducted a retrospective study with cancer register data to investigate associations between body weight, PA, and cancer risk among Finnish LS carriers. The participants (*n* = 465, 54% women) self-reported their adulthood body weight and PA at 10-year intervals. Overall cancer risk and colorectal cancer (CRC) risk was analyzed separately for men and women with respect to longitudinal and near-term changes in body weight and PA using extended Cox regression models. The longitudinal weight change was associated with an increased risk of all cancers (HR 1.02, 95% CI 1.00–1.04) and CRC (HR 1.03, 1.01–1.05) in men. The near-term weight change was associated with a lower CRC risk in women (HR 0.96, 0.92–0.99). Furthermore, 77.6% of the participants retained their PA category over time. Men in the high-activity group had a reduced longitudinal cancer risk of 63% (HR 0.37, 0.15–0.98) compared to men in the low-activity group. PA in adulthood was not associated with cancer risk among women. These results emphasize the role of weight maintenance and high-intensity PA throughout the lifespan in cancer prevention, particularly in men with LS.

## 1. Introduction

Colorectal cancer (CRC) is the third most commonly diagnosed cancer in Europe, with an estimated 499,667 new cases in 2018, and is the second most common cause of cancer mortality [1]. Approximately 3–5% of all CRC cases may be due to hereditary cancer syndrome, also known as Lynch syndrome (LS) [2], which is caused by pathogenic germline variants in DNA mismatch repair genes (*path_MMR*): *MLH1*, *MSH2*, *MSH6*, or *PMS2* [3]. Individuals with *path_MMR* variants are at considerably greater risk of developing CRC (40–80%), endometrial cancer (40–60%) and various other cancers compared to the general population [4,5]. However, cancer risk is highly variable among different mutation carriers and thus far it is not known why not all *path_MMR* carriers develop cancer, whereas others develop cancer at a young age and/or suffer from multiple different types of cancers during their lifespan.

There is strong evidence derived from the general population that increased physical activity and reduced body adiposity are associated with decreased cancer risk [6,7,8,9,10,11,12]. Recent research suggests that LS CRC risk can also be moderated by these lifestyle factors [13,14,15,16,17], but the number of studies that investigate the associations between lifestyle and LS cancer risk are scarce. Currently, there are only two studies that we are aware of that have assessed the association of physical activity and LS cancer risk [13,14] and they have not taken into consideration the fact that this potential association may vary during a lifespan. In addition, the evidence regarding the associations of cancer risk and obesity and overweight is contradictory among women with LS [15], and thus, the risk analyses should be performed separately for both sexes. It is of great importance to identify modifiable behavioural factors of LS cancer risk to motivate variant carriers to change suboptimal conduct or maintain a healthy lifestyle. Lifestyle modification could efficiently aid in reducing individual cancer risk despite a strong genetic predisposition.

In this study, we hypothesized that the interplay between genetic factors and lifestyle is associated with variable cancer risk in a distinct high-risk population. Two founder mutations in *MLH1* are found in a major proportion of Finnish LS families [18,19,20], which offers a possibility for investigating lifestyle factors as a modifier of cancer risk in a relatively homogenous LS population and hence may limit the influence of genetic discrepancies. The aim of this retrospective study with longitudinal lifestyle and cancer register data was to investigate associations between body weight and physical activity on CRC and overall cancer risk among adult Finnish *path_MMR* men and women. We modeled the recalled levels of weight and physical activity as time-dependent variables in the relative risk model.

## 2. Results

### 2.1. Descriptive Statistics

Descriptive data on *path_MMR* variants, cancer history, and lifestyle and socioeconomic characteristics are presented in Table 1. Of the 465 participants, 215 (46.2%) were men and 250 (53.8%) were women. The mean age at the time of data collection was 56.4 years vs. 57.4 years, respectively.

Almost half of the men (47.0%) and women (49.0%) had had one or more cancers prior to lifestyle data collection. The most common cancer was CRC, with a higher prevalence in men (88.1%) compared to women (63.1%). The mean age at the first cancer incidence was 45.9 years in men and 48.6 years in women, and 45.6 years and 48.4 years at the age of the first CRC, respectively. MMR-gene variant frequencies for the entire study population were 47.3% for *MLH1*, 25.6% for *MLH1* other than ex 16, 16.3% for *MSH2*, 10.3% for *MSH6*, and 0.4% for *PMS2*.

Healthy participants were more often working, whereas participants with cancer were more often retired. Women with cancer had a lower level of education and they were more commonly living alone compared to their healthy counterparts. Self-rated health status and fitness, as well as current physical activity level, was in general better among participants who had not had cancer. In addition, other lifestyle variables were moderately similar among participants with and without cancer.

Table 2 describes the cumulative cancer event history of the entire study population during retrospective follow-up separately for both sexes. Among both men and women, most of the cancer events occurred from the age of 40 years to 70 years.

### 2.2. Body Weight History

Table 3 describes the changes in mean body weight during the retrospective follow-up. The mean body weight increased throughout the lifespan in both sexes. Furthermore, the mean individual change in body weight was positive in each 10-year interval, both among men and women during the adult lifespan. From the age of 40 years onwards, average individual weight increased with respect to recalled weight at the age of 20 years and ranged between 8 kg and 12 kg per year for men and between 8 kg and 13 kg for women.

Table 4 presents the associations between adult life body weight and cancer risk. For consistency, all results in the text regarding body weight and cancer risk are presented only from models adjusted for height, MMR gene, education, alcohol consumption, smoking status, and the use of anti-inflammatory drugs.

#### 2.2.1. Risk of All Cancers

A change in longitudinal weight throughout the lifespan, calculated per one-kg weight increase, was associated with a 2% increased risk of cancers in men (HR 1.02, 95% CI 1.00–1.03), whereas no such association was observed in women. Moreover, near-term weight change had no impact on the risk of all cancers in either sex.

#### 2.2.2. Risk of CRC

Longitudinal weight gain increased the CRC risk by 3% in men only (HR 1.03, 95% CI 1.01–1.05). Among women, near-term weight gain within the 10-year interval before cancer diagnosis was associated with a 4% decreased risk of CRC (HR 0.96, 95% CI 0.92–0.99). No associations between near-term weight gain and CRC risk were observed in men.

### 2.3. Physical Activity during Adulthood

Table 5 describes changes in the intensity of physical activity during the retrospective follow-up. In both sexes, a great majority retained their activity category over time. However, when the category changed, it was more common among men to move from the organized physical activity participation category to the lower activity category.

The associations between physical activity and cancer risks are presented in Table 6. For consistency, all results in the text regarding physical activity and cancer risk are presented only from adjusted models.

#### 2.3.1. Risk of All Cancers

Men in the high activity group were found to have a reduced longitudinal cancer risk of 63% (HR 0.37, 95% CI 0.15–0.98) compared to men in the low activity group. There were no longitudinal associations between physical activity and cancer risk observed in women. In the near-term, participating in physical activity had no impact on cancer risk in either men or women.

#### 2.3.2. Risk of CRC

There was no association between physical activity and the risk of CRC.

## 3. Discussion

We conducted a retrospective study with longitudinal data collection and cancer register data among Finnish *path_MMR* carriers to elucidate the associations between changes in adult body weight, physical activity, and cancer risk. Our results suggest that associations between lifestyle and cancer risk differ between men and women and may vary during the course of life. We found that an overall increase in total body weight throughout the lifespan slightly elevated the risk of cancers, including CRC, in men. We also observed that men who continued to participate in more intensive physical activities over their adult life were at lower risk of all cancers.

In Western societies, body weight typically accumulates during the adult lifespan and growing levels of obesity predispose individuals to multiple health complications. Obesity is acknowledged as one of the most important risk factors of non-communicable diseases [9,21,22]. It is well established in the general population that excess body weight and adiposity, particularly in overweight individuals, is an important risk factor for several cancers [8] and the risk could be reduced by lowering excess body mass [23]. In the population of *path_MMR* carriers of the current study, participants increased their body weight during their adult years, and 61% of men and 55% of women were overweight or obese. Obesity and overweight may be more harmful for men, as we found that a trend of body weight accumulation during adult years was associated with an increased cancer risk in men but not in women. This is in agreement with other reports which have investigated weight accumulation in relation to the risk of LS cancer [17,24].

Unfortunately, in most of the studies, including ours, a lifelong change in body composition could only be determined by changes in body weight. However, aging is not only associated with an increase in body weight that is related to fat accumulation but also to changes in body composition [25]. Beginning from the age of 30 years, muscle mass tends to decrease and the decline accelerates after the age of 50 years, particularly among women due to menopause [26]. Concurrently, the amount and distribution of body fat may change, thereby resulting in the accumulation of fat—particularly in visceral areas—which increases cancer risk through several already identified biological pathways. The best-characterized association is between abdominal obesity and disturbed insulin metabolism, which may influence cancer risk through cell proliferation and apoptosis [10,27]. Age-related trends in body weight accumulation are different between men and women, which may explain our dissimilar findings regarding weight accumulation and cancer risk. In general, men are more prone to increased android-type fat distribution—that is, abdominal fat distribution—throughout their lifespan [28]. In contrast, women tend to be more prone to gynoid-type fat distribution during their premenopausal years and then shift to androgen-type fat accumulation in their postmenopausal years [29]. In our study, the weight accumulation was lower among women, but we cannot exclude the potential confounding role of menopause on the association between body weight and cancer risk as our data did not include information regarding the menopausal status of the female participants. Based on the population averages of menopause age being between 50 years and 53 years, we can estimate that 67% of women participating in this study were post-menopausal, but the considerable individual variation observed makes this estimate imprecise [30]. Nevertheless, visceral adiposity has been associated with higher cancer risk in both sexes, although in women the risk estimates appear to differ between pre- and post-menopausal women [31].

Intriguingly, we also found that near-term weight increase had a CRC-protective effect in women. To speculate, it is possible that hormonal factors might have influenced the risk estimates. The primary source of endogenous estrogen in post-menopausal women has been suggested to be adipose tissue [32,33]. Therefore, higher adiposity could maintain a higher systemic estrogen level, which in turn may provide some protection through (for instance) the anti-inflammatory action of estrogens [34]. However, this is highly speculative. There is evidence that exogenous estrogen use can be cancer protective, neutral or to increase the risk of different cancers [35,36,37]. In the current study, we did not investigate the potential role of hormone therapy, nor did we have the ability to measure estrogen levels; thus, we cannot exclude the role of systemic estrogen level. Therefore, as the great majority of CRC-diagnosed women in our study were over 60 years of age, there remains a possibility that the plausible protective effects of estrogen derived from adipose tissue might have masked the effect of weight gain on colorectal cancer risk.

Our results suggest that performing more vigorous guided physical activity exhibit a cancer-preventive effect in men, as those who continued at higher levels of physical activity were at a 63% lower cancer risk when compared to less active men performing non-guided physical activity. To date, we are aware of only two previous studies that have assessed the impact of physical activity on LS cancer risk [13,14]. Both designs were retrospective like our study but did not assess risk estimates separately for men and women. Kamiza et al. (2015) [13] reported that among 301 Taiwanese individuals (51.8% women) carrying *path_MLH1* and *path_MSH2* regular vigorous leisure time physical activity over a year prior to cancer diagnosis decreased CRC risk by 38% when compared to those who did not indulge in any such activity. Although we did not observe associations between physical activity and cancer risk in the near-term like Kamiza et al. (2015) [13], similarly to their findings, our results also suggested that performing vigorous physical activity could reduce cancer risk.

Further, the study by Dashti et al. (2018) [14] comprised 2042 *path_MMR* carriers (57% women). As in our study, they modeled longitudinal and near-term changes in physical activity separately. Unlike our study, Dashti et al. (2018) [14] did not find an association between physical activity and cancer risk when assessed over several age periods, even though a trend of lowering the risk of CRC was observed. In addition, they used MET-h/week to assess the amount of physical activity, but we did not do so. Near-term cancer risk assessment, which was used in the current study, may be particularly effective for identifying specific suboptimal lifestyle changes that could be associated with carcinogenesis and precede cancer occurrence. Dashti et al. (2018) [14] found higher levels of near-term physical activity (>35 MET-h/week) to be protective against such a risk, whereas we did not find an association between near-term physical activity and cancer risk (any cancer or CRC).

We did not find longitudinal or near-term evidence linking physical activity to cancer risk in *path_MMR* women, which may be due to differences in physical activity behavior between men and women, including the intensity and type of physical activity, as well as the timing of physical activity exposure in life. For example, throughout the follow-up, female participants mainly reported participating more frequently in guided leisure time physical activity than men, who reported performing competitive sports more often (Appendix A). Overall, our results highlight the potential role of physical activity in cancer prevention among *path_MMR* carrier men, as already advocated in clinical guidelines for LS [38]. Although most *path_MMR* carriers suffer from cancer at some point of their life, this is an important finding.

Various reports in the extant literature have suggested several mechanisms that link physical activity with a reduced cancer risk [11]. For example, physical activity produces multiple beneficial changes in cardiorespiratory systems [39], and being physically active also helps with weight control, as well as with reducing excess adiposity [12]. These combined effects might have a beneficial impact on biological mechanisms that interact directly or indirectly with cancer, such as improved insulin sensitivity and reduced chronic low-level inflammation, which is also linked with favorable immunomodulation [40,41]. However, the existing evidence originates from sporadic cancer patients who could differ from *path_MMR* carriers with respect to disease mechanisms, carcinogenesis, and biological regulation.

As described previously, the association of decreased cancer risk and healthy lifestyle in the general population has been observed in large population-based studies. Since the influence of healthy behaviour on decreased cancer risk was observed in our study with a limited number of participants, it could be possible that the effect of the modifiable behavioral risk factors—physical activity and body weight—is emphasised in *path_MMR* carriers due to their strong genetic predisposition to cancers. Therefore, it is important to follow these modifiable risk factors among *path_MMR* carriers during their regular healthcare visits. An optimal lifestyle could partially compensate for the strong genetic predisposition to cancers and thus help in cancer prevention. Nowadays, individual cancer risk can be calculated and demonstrated via online tools (www.plsd.eu, accessed 9 January 2021), which can be used to improve motivational support for healthy lifestyle maintenance or lifestyle changes.

A major strength of the current study is that the study cohort comprised participants who had undergone comprehensive screenings of LS-predisposing mutations, with ascertainment utilizing Amsterdam and Bethesda clinical criteria and cascade testing, and those who had been offered colonoscopy surveillance at 2–3-year intervals. Our body weight and physical activity data collection encompassed the entire adult lifespan and was carefully analyzed by considering potential time-varying risks, sex differences, and potential confounders regarding cancer incidence. The observation period was initiated from the age of 20 years, instead of birth, to avoid the detection of changes in body weight which were merely due to natural growth and maturation. In doing so, we were also able to exclude the time-period when cancer incidence tends to be extremely low even among *path_MMR* carriers. We also chose to model the change in cancer risk in the near-term setting (during the age-period of cancer or censoring) as it could be more accurate than longitudinal change, which could be influenced by poor recall. We also used time-dependent covariate values, which allow us to account for changes in predictor values over time.

However, there are also several limitations. Weight and physical activity were assessed using self-recall instruments. Even though a recent study found that cross-sectional self-reported measurements of BMI were reasonably close to recent direct measurements [42], the recall of weight in the more distant past has lower reliability [43] and for some of our older participants, the recall time was several decades. Moreover, there might also be sex-based discrepancies, as women tend to underestimate their weight and men tend to overestimate it [44]. Finally, Smith et al. (2013) [45] found the recall of physical activity of the distant past to be moderately reproducible, but poor at the individual level. Taken together, we cannot exclude the possibility that recall bias might have influenced the risk estimates.

## 4. Materials and Methods

### 4.1. Study Sample

The study cohort included those carriers of *path_MMR* who were registered in the Finnish Lynch Syndrome Research Registry (LSRFi; www.lynchsyndrooma.fi, accessed 16 June 2020) and provided consent for research-related contacts. LSRFi is a nation-wide research registry (est. 1982) operating in Jyväskylä and Helsinki that organizes surveillance and cancer prevention for LS families. Currently, the registry consists of clinical and family history data of over 300 LS families and over 1700 pathogenic variant carriers under frequent surveillance. Individuals were identified in the registry before the genetic testing became available, based on clinical criteria (Amsterdam and Bethesda criteria) [46,47], and subsequently through cascade testing of the families and universal testing of tumors. Adult members of LSRFi with confirmed *path_MMR* variants (classes 4 and 5 by InSiGHT criteria) [48] were eligible for the study.

### 4.2. Cancer Register Data

Age, sex, and all cancer diagnoses with the cancer type and date of each diagnosis, mutation status, and family cancer history were confirmed from hospital medical records and national cancer registries upon recording in the LSRFi. With regard to analyses, participants in the cancer group were required to have at least one past cancer diagnosis in the medical registries although he/she could have been healthy at the time of data collection. The healthy group included only *path_MMR* carriers who had remained cancer free until data collection.

### 4.3. Questionnaire Data Collection

Questionnaires for anthropometric, socioeconomic, and lifestyle data collection were sent to 1038 adult *path_MMR* carriers whose addresses were available in LSRFi in December 2016 and July 2020. Of them, 480 (response rate 46.2%) returned the questionnaire. However, 15 participants did not carry the *path_MMR* variant and therefore they did not fulfil the eligibility criteria and were excluded from the study. Then, the final study sample included 465 participants.

### 4.4. Descriptive Variables

#### 4.4.1. Socioeconomic Characteristics

The education level was categorized according to the Finnish schooling system into four categories: basic education (including elementary or comprehensive school), upper secondary education (including vocational school and high school level degrees), polytechnic degree, and university degree. Occupational status included the categories of worker or employee, retiree (including both disability and old age pensioners), and other (including students, unemployed people, and people on parental leave). Marital status was categorized as living alone or married/cohabitating.

#### 4.4.2. Perceived Health and Physical Fitness

Self-rated health and physical fitness were collected using standard five-scale questions. Due to the low number of responses in a few categories, answers were re-categorized into poor (also including very poor), average, and good (also including very good) for statistical analysis.

#### 4.4.3. Lifestyle Variables

The level of alcohol consumption was identified by two questions assessing the frequency of alcohol use and the number of alcohol portions consumed per occasion. One portion refers to 10–14 g alcohol, which one may obtain, for example, from a single serving of 0.33 liters of beer or a similar light alcoholic beverage, from 12 cL of wine, or from 4 cL of spirits. Furthermore, the subjects’ smoking status was defined as never smoker if they reported being non-smokers and having never been a smoker, or smoked <100 cigarettes during their entire life; as former smoker if they reported currently being non-smokers but were regular smokers in the past; or as current smoker if they reported being current and regular smokers. Participants were also asked whether or not they used any anti-inflammatory drugs regularly during the surveyed time period (yes/no). Current leisure-time physical activity was assessed via the seven-option scale question [49,50]. The scale options were re-categorized into low (light walking and outdoor activities 1–2 times per week), medium (some light walking and outdoor activities several times a week, or engaging in brisk physical activity 1–2 times per week causing some shortness of breath and perspiration), and high (brisk physical activity 3–5 times a week causing some shortness of breath and perspiration or fitness training several times a week causing heavy perspiration and being out-of-breath during exercise or playing competitive sports and maintaining regular fitness).

#### 4.4.4. Anthropometrics

Anthropometrics were self-measured. Participants were asked to report their height and to measure their body weight before breakfast and without clothes. If weight could not be measured, participants were asked to fill in the last weight measurement known. BMI was calculated as weight in kilograms divided by the height squared in meters (kg/m^2^). Participants were categorized into four BMI groups—underweight (BMI < 18.5), normal weight (BMI 18.5–24.9), overweight (BMI 25–29.9), and obese (BMI ≥ 30)—according to the WHO classifications. Measuring tape was sent with the questionnaire to examine the waist circumference, along with written instructions. Waist circumference was measured without clothing in a standing position, 2 cm above the umbilicus.

### 4.5. Outcome Variables for Life-Long Exposures

To measure body weight history during adulthood, participants recalled their body weight in kilograms at age 20, 30, 40, 50, 60, and up to 70+ years.

Physical activity during adulthood was assessed via four-option scale questions through which participants recalled the level of regular physical activity they had at different adult age ranges throughout their lives [51]. Participants reported their past physical activity at age ranges 20–29, 30–39, 40–49, 50–59, 60–75, and 75+ years up to their current age period at the time of measurement. The four response options for each age-period were (1) no regular physical activity, (2) regular independent leisure-time physical activity (all non-organized occupational or leisure-time physical activity, i.e., commuting, school/work activities), (3) regular goal-oriented competitive sport and training related to that sport, and (4) other regular supervised physical activity (physical activity that was organized in a sport club, etc., but was not related to competitive sports participation). These four categories were re-categorized into low activity (options 1 and 2) and high activity (options 3 and 4). The same re-categorization was applied to each age range.

### 4.6. Statistical Analysis

Means and standard deviations were used as descriptive statistics for continuous measurements, frequencies, and percentages of categorical data. The proportional hazards model, extended for time-dependent covariates, was used in modeling the association of weight and physical activity on cancer incidence in (1) longitudinal and (2) near-term settings. We used age as the time variable and determined cancer status at the end of follow-up. Thus, follow-up time extended from study entry at the age of 20 years to exit-age due to cancer diagnosis (event) or remaining free of cancer (censored). As we used time-dependent measurements of weight and physical activity, we utilized the counting process approach [52] for the analysis of the relative risk of cancer related to these exposures. Data were divided into 10-year intervals and each interval was represented by the weight and physical activity measurements of that interval. Due to sex-related differences, we reported models separately for men and women and we also constructed separate models for cancers of any type and CRC; we also used a joint weight/physical activity model to examine the possible interaction between these two exposures. Further details of the models, data management, and model diagnostics can be found in the Appendix A. We reported hazard ratios from the crude unadjusted model, as well as a model adjusted for the affected MMR-gene variant, height, education, smoking, alcohol use, and the use of anti-inflammatory medication. Nested random effects were used to adjust for individuals within the family structure.

## 5. Conclusions

To conclude, our results suggest that men with *path_MMR* were particularly susceptible to lifestyle exposures that may be either protective or hazard increasing. We found that weight gain in adulthood increased the risk of cancer in men, whereas participating in more intense physical activity across the lifespan may have a cancer-preventive effect. According to our results, women were not as prone to lifestyle-related risk factors. The sex-based difference in the associations could be explained by differences in weight gain, which was smaller in women, and by sex-related factors modifying body composition over time. Taken together, our results emphasize the importance of weight maintenance and high-intensity physical activity throughout the lifespan in cancer prevention in men with *path_MMR*. The results of our study could be used in developing a cancer risk quantification methodology based on the consideration of various risk factors that are modifiable by the individuals themselves.

## Figures and Tables

**Table 1 cancers-13-01849-t001:** Descriptive demographic, genetic, cancer history, socioeconomic, and lifestyle-related characteristics of the study population.

Background Variable	Men		Women	
	Cancer	Healthy	Cancer	Healthy
	*N* = 101	*N* = 114	*N* = 122	*N* = 128
Age at data collection (years (SD))	64.2 (11.2)	49.5 (14.0)	65.4 (9.6)	49.7 (14.0)
Age at first cancer (years (SD))	45.9 (10.4)		48.6 (29–79)	
Age at first CRC (years (SD))	45.6 (11.1)		48.4 (29–79)	
Cancers diagnosed (*n* (%))				
CRC	89 (88.1)		77 (63.1)	
Endometrial cancer	17 (16.8)		57 (46.7)	
Other cancers ^a^			30 (24.6)	
MMR gene affected (*n* (%))				
*MLH1*	50 (49.5)	56 (49.1)	50 (41.0)	64 (50.0)
*MLH1* other than exon 16 deletion	24 (23.8)	26 (22.8)	35 (28.7)	34 (26.6)
*MSH2*	15 (14.9)	22 (19.3)	26 (21.3)	13 (10.2)
*MSH6*	10 (9.9)	10 (8.8)	11 (9.0)	17 (13.3)
*PMS2*	2 (2.0)			
Socioeconomic characteristics				
Education (*n* (%))				
Basic education	20 (19.8)	13 (11.4)	36 (29.5)	16 (12.5)
Upper secondary degrees	46 (45.6)	60 (52.6)	51 (41.8)	63 (49.4)
Polytechnic degree	13 (12.9)	18 (15.8)	11 (9.0)	19 (14.8)
University degree	22 (21.8)	23 (20.2)	24 (19.7)	30 (23.4)
Occupational status (*n* (%)) ^#^				
Worker/employee	40 (39.6)	76 (66.7)	47 (39.2)	92 (72.4)
Retired	45 (44.6)	18 (15.8)	56 (46.7)	22 (17.3)
Other ^b^	16 (15.8)	20 (17.5)	17 (14.2)	13 (10.3)
Marital status (*n* (%))				
Living alone	20 (19.8)	22 (19.3)	40 (32.8)	32 (25.0)
Married/cohabitation	81 (80.2)	92 (80.7)	82 (67.2)	96 (75.0)
Perceived health and physical fitness				
Self-rated health (*n* (%))				
Poor	16 (15.8)	13 (11.4)	15 (12.3)	19 (14.8)
Average	39 (38.6)	27 (23.7)	49 (40.2)	32 (25.0)
Good	46 (45.5)	74 (64.9)	57 (46.7)	77 (60.2)
Self-rated physical fitness (*n* (%))				
Poor	22 (21.8)	18 (15.8)	16 (13.1)	21 (16.4)
Average	39 (38.6)	29 (25.4)	50 (41.0)	40 (31.3)
Good	40 (39.6)	67 (58.8)	56 (45.9)	67 (52.3)
Lifestyle variables				
Alcohol consumption (portions/week (SD)) ^#^	4.8 (6.5)	4.8 (5.3)	2.3 (3.5)	3.0 (4.3)
Smoking status (*n* (%)) ^#^				
Never	43 (42.6)	45 (39.5)	61 (50.0)	64 (50.0)
Former	47 (46.5)	48 (42.1)	52 (42.6)	46 (35.9)
Current	11 (10.9)	21 (18.4)	8 (6.6)	17 (13.3)
Use of anti-inflammatory drugs (*n* (%)) ^#^				
No	83 (85.6)	89 (78.1)	88 (72.7)	84 (65.6)
Yes	14 (14.4)	25 (21.9)	33 (27.3)	44 (34.4)
Current physical activity (*n* (%)) ^#^				
Low	28 (28.0)	23 (20.2)	23 (18.9)	21 (16.4)
Medium	32 (32.0)	24 (21.1)	44 (36.1)	47 (36.7)
High	40 (40.0)	67 (58.8)	55 (45.1)	60 (46.9)
BMI (kg/m^2^ (SD)) ^#^	27.2 (5.3)	26.6 (4.2)	27.1 (5.8)	27.6 (11.6)
BMI categories (*n* (%)) ^#^				
Underweight	2 (2.0)	2 (1.8)	1 (0.8)	2 (1.6)
Normal weight	40 (40.0)	38 (33.3)	48 (40.7)	56 (44.1)
Overweight	34 (34.0)	57 (50.0)	39 (33.1)	34 (26.8)
Obese	24 (24.0)	17 (14.9)	30 (25.4)	35 (27.6)
Waist circumference (cm (SD)) ^#^	100.5 (14.4)	97.8 (11.0)	90.6 (14.4)	88.6 (14.5)

^#^ Missing data: occupational status *n* = 3, alcohol consumption *n* = 3, smoking status *n* = 2, anti-inflammatory drugs *n* = 5, BMI and BMI categories *n* = 6, current physical activity *n* = 1, waist circumference *n* = 8. ^a^ other cancers included breast cancer, ovarian cancer, prostate cancer and skin cancers; ^b^ other included students, unemployed, and persons on parental leave. BMI = body mass index; CRC = colorectal cancer.

**Table 2 cancers-13-01849-t002:** Summary of individuals at risk for cancer, events, and censorings occurring at the beginning of 10-year periods for men and women.

Sex	Period	At Risk	All Cancers		CRC	
			Events	Censored	Events	Censored
Men						
	(20, 30)	215	0	0	0	0
	(30, 40)	199	5	11	5	11
	(40, 50)	156	30	29	25	34
	(50, 60)	89	65	61	55	71
	(60, 70)	48	83	84	69	98
	(70, 77)	8	98	109	80	127
	77	0	98	117	80	135
Women						
	(20, 30)	250	0	0	0	0
	(30, 40)	242	1	7	1	7
	(40, 50)	190	25	35	19	41
	(50, 60)	117	67	66	40	93
	(60, 70)	55	100	95	55	140
	(70, 80)	14	117	119	63	173
	(80, 85)	1	120	129	65	184
	85	0	120	130	65	185

Period: A 10-year interval (years). At risk: participants at risk (*n*) at each 10-year interval. Events: cumulative *n* of occurred cancer events. Censored: cumulative *n* of censored participants. CRC = colorectal cancer. Note: for time interval lower limit includes the indicated value and the value for the upper limit is excluded.

**Table 3 cancers-13-01849-t003:** Means and standard deviations for weight measurements (kg) and individual weight change in 10-year periods among participants with Lynch syndrome.

Sex	*N*	Period	At Risk, Start of Period		Individual Change, Near-Term		Individual Change, Longitudinal	
			Mean	SD	Mean	SD	Mean	SD
Men								
	215	(20, 30)	72.65	11.93	-	-	-	-
	199	(30, 40)	77.06	12.68	5.06	5.99	5.06	5.99
	156	(40, 50)	79.56	14.91	3.76	7.05	8.23	10.29
	89	(50, 60)	79.67	12.43	4.03	6.73	10.61	10.52
	48	(60, 70)	77.05	11.76	0.72	4.10	8.22	9.25
	8	(70, 77)	83.40	15.37	4.60	7.09	12.20	12.28
Women								
	250	(20, 30)	59.16	11.28	-	-	-	-
	242	(30, 40)	62.88	13.48	3.96	6.59	3.96	6.59
	190	(40, 50)	66.30	14.78	3.98	5.86	7.53	8.72
	117	(50, 60)	68.30	13.65	2.83	6.11	9.33	10.29
	55	(60, 70)	68.98	11.58	2.85	4.41	9.39	9.24
	14	(70, 80)	73.91	12.84	4.73	9.02	13.18	11.97
	1	(80, 85)	79.00	- ^a^	0.00	- ^a^	20.00	- ^a^

^a^ The variation is estimable only for one case. Period: A 10-year interval (years). Longitudinal change: change in body weight (kg) relative to the body weight at the age of 20 years. Near-term change: change in body weight (kg) relative to the body weight at the previous 10-year interval before diagnosis or censoring. Mean (kg). SD = standard deviation (kg). Note: for time interval lower limit includes the indicated value and the value for the upper limit is excluded.

**Table 4 cancers-13-01849-t004:** Hazard ratios for associations of body weight and cancer risk in participants with Lynch syndrome.

Setting	Unadjusted Model				Adjusted Model *			
	Cancer Events	Observations	HR (95% CI)	*p*-Value	Cancer Events	Observations	HR (95% CI)	*p*-Value
Men								
All cancers								
Longitudinal change	77	610	1.02 (1.00–1.03)	0.048	74	579	1.02 (1.00–1.04)	0.022
Near-term change	77	185	0.99 (0.97–1.00)	0.345	74	174	0.99 (0.97–1.01)	0.424
Colorectal cancer								
Longitudinal change	60	610	1.02 (1.00–1.03)	0.023	57	579	1.03 (1.01–1.05)	0.004
Near-term change	60	185	1.00 (0.98–1.01)	0.695	57	174	1.00 (0.98–1.02)	0.861
Women								
All cancers								
Longitudinal change	95	758	0.99 (0.97–1.00)	0.290	91	720	1.00 (0.98–1.02)	0.887
Near-term change	95	221	0.98 (0.97–1.00)	0.059	91	209	0.98 (0.96–1.00)	0.059
Colorectal cancer								
Longitudinal change	50	758	0.99 (0.96–1.01)	0.258	48	720	0.99 (0.96–1.02)	0.454
Near-term change	50	221	0.98 (0.95–1.00)	0.106	48	209	0.96 (0.92–0.99)	0.015

* Model adjusted for height, MMR-gene, education, smoking, alcohol consumption, and use of anti-inflammatory drugs. Longitudinal change: longitudinal change in body weight from the age of 20 years until the first cancer diagnosis or censoring. Near-term change: age-stratified change in body weight relative to the body weight at the previous 10-year interval before diagnosis or censoring. *p*-values statistically significant at <0.05 level. Statistically significant hazard ratios are highlighted in bold. Cancer events: number of occurred cancers. Observations: number of observations across each 10-year interval. HR = hazard ratio. CI = confidence interval.

**Table 5 cancers-13-01849-t005:** Frequencies and cross-period tables for participation in organized physical activity in 10-year periods among participants with Lynch syndrome.

Sex	Period	Physical Activity	*N* (%)	Near-Term Change		Longitudinal Change	
				Low Activity	High Activity	Low Activity	High Activity
Men	(20, 30)	Low activity	155 (76)	-	-	-	-
		High activity	50 (24)	-	-	-	-
	(30, 40)	Low activity	153 (81)	131 (94)	22 (44)	131 (94)	22 (44)
		High activity	37 (19)	9 (6)	28 (54)	9 (6)	28 (56)
	(40, 50)	Low activity	129 (86)	121 (98)	8 (30)	104 (95)	25 (62)
		High activity	21 (14)	2 (2)	19 (70)	6 (5)	15 (38)
	(50, 60)	Low activity	77 (91)	70 (100)	7 (47)	59 (97)	18 (75)
		High activity	8 (9)	0 (0)	8 (53)	2 (3)	6 (25)
	(60, 70)	Low activity	43 (96)	42 (100)	1 (33)	31 (100)	12 (86)
		High activity	2 (4)	0 (0)	2 (67)	0 (0)	2 (14)
	(70, 77)	Low activity	8 (100)	8 (100)	0 (-)	6 (100)	2 (100)
		High activity	0 (0)	0 (0)	0 (-)	0 (0)	0 (0)
Women	(20, 30)	Low activity	180 (76)	-	-	-	-
		High activity	58 (24)	-	-	-	-
	(30, 40)	Low activity	166 (72)	151 (87)	15 (26)	151 (87)	15 (26)
		High activity	64 (28)	22 (13)	42 (74)	22 (13)	42 (74)
	(40, 50)	Low activity	121 (68)	114 (87)	7 (15)	111 (92)	31 (53)
		High activity	58 (32)	17 (13)	41 (85)	10 (8)	27 (47)
	(50, 60)	Low activity	86 (77)	75 (96)	11 (32)	82 (95)	13 (50)
		High activity	26 (23)	3 (4)	23 (68)	4 (5)	13 (50)
	(60, 70)	Low activity	40 (78)	38 (97)	2 (17)	38 (95)	5 (45)
		High activity	11 (22)	1 (3)	10 (83)	2 (5)	6 (55)
	(70, 77)	Low activity	10 (83)	10 (100)	0 (0)	10 (100)	1 (50)
		High activity	2 (17)	0 (0)	2 (100)	0 (0)	1 (50)

Period: A 10-year interval (years). Physical activity: physical activity group at the beginning of the 10-year interval. Longitudinal change: change in physical activity group (*n*) relative to the physical activity group at the age of 20 years. Near-term change: change in physical activity group (*n*) relative to the physical activity group at the previous 10-year interval. Note: for time interval lower limit includes the indicated value and the value for the upper limit is excluded.

**Table 6 cancers-13-01849-t006:** Hazard ratios for associations of physical activity and cancer risk in participants with Lynch syndrome.

Setting	Unadjusted Model				Adjusted Model *			
	Cancer Events	Observations	HR (95% CI)	*p*-Value	Cancer Events	Observations	HR (95% CI)	*p*-Value
Men								
All cancers								
Longitudinal change	91	683	0.44 (0.19–1.04)	0.063	86	648	0.37 (0.15–0.98)	0.044
Near-term change	91	205	0.69 (0.29–1.64)	0.403	86	192	0.74 (0.27–2.01)	0.557
Colorectal cancer								
Longitudinal change	73	683	0.57 (0.25–1.33)	0.194	68	648	0.52 (0.20–1.36)	0.181
Near-term change	73	205	0.93 (0.39–2.24)	0.874	68	192	0.99 (0.36–2.73)	0.983
Women								
All cancers								
Longitudinal change	110	823	1.31 (0.86–1.97)	0.206	107	789	1.26 (0.79–2.00)	0.341
Near-term change	110	238	1.42 (0.89–2.25)	0.138	107	227	1.34 (0.80–2.23)	0.268
Colorectal cancer								
Longitudinal change	59	823	1.16 (0.65–2.10)	0.612	57	789	1.28 (0.65–2.52)	0.471
Near-term change	59	238	0.92 (0.49–1.72)	0.797	57	227	0.99 (0.48–2.02)	0.973

* Model adjusted for MMR-gene, education, smoking, alcohol consumption, and use of anti-inflammatory drugs. In all analyses, the reference group was *Low activity.* Longitudinal change: longitudinal change in physical activity from the age of 20 years until the first cancer diagnosis or censoring. Near-term change: age-stratified change in physical activity level relative to the physical activity level at the previous 10-year interval before diagnosis or censoring. *p*-values statistically significant at the <0.05 level. Statistically significant hazard ratios are highlighted in bold. Cancer events: number of occurred cancers. Observations: number of observations across each 10-year interval. HR = hazard ratio. CI = confidence interval.

## Data Availability

The data are not publicly available due to privacy or ethical restrictions and EU legislation. However, application for the clinical datasets can be made via The Finnish Lynch Syndrome Research Registry and for the datasets obtained through questionnaire survey via contacting the corresponding author.

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
