# Peer review of "Body Weight, Physical Activity, and Risk of Cancer in Lynch Syndrome"

_cancers, 2021, doi:10.3390/cancers13081849_

Round 1

Reviewer 1 Report

The authors examined the associations between body weight, physical activity, and colorectal cancer risk among patients with Lynch syndrome. They found that weight gain increased cancer risk, while being highly physically active during adulthood could decreased cancer risk only in men. However, there are several concerns in this study.

Major points

  1. The aim of this study was to examine the associations between body weight, physical activity and colorectal cancer. Therefore, it is necessary to collect the Lynch syndrome patients with colorectal cancer and Lynch syndrome patients without colorectal cancer, then they should be compared.
  2. The systematic review have already reported that increased physical activity and lower body adiposity are also associated with decreased risk of cancers. Therefore, the findings in this study are not novel.
  3. In Table 1, average ages were different among two group. Therefore, physical activity level can be different among them.
  4. The number of subjects with 60-77 years is too small to conclude.
  5. It is not clear how healthy group was extracted.
  6. The information of body weight, physical activity was collected by re-call. Therefore, recall bias should be considered.

7. Overall, the findings in this study are not enough for the level of this journal.

Author Response

Reviewer #1

The authors examined the associations between body weight, physical activity, and colorectal cancer risk among patients with Lynch syndrome. They found that weight gain increased cancer risk, while being highly physically active during adulthood could decreased cancer risk only in men. However, there are several concerns in this study.

Reviewer comment 1:

The aim of this study was to examine the associations between body weight, physical activity and colorectal cancer. Therefore, it is necessary to collect the Lynch syndrome patients with colorectal cancer and Lynch syndrome patients without colorectal cancer, then they should be compared.

Author response:

We thank the Reviewer for the feedback and apologize that the aim of the study was stated unclearly. The aim of our study was to examine whether lifelong changes in body weight and physical activity are associated with Lynch syndrome (LS) cancer incidence from age 20 onwards, including but not limited to, colorectal cancer. The cohort included both healthy participants and participants with cancer(s). The statistical analyses were rather complex, and are extensively described in the supplement file. The main point in our analysis was time-dependency. We used the counting process data format to model the recalled levels of weight and physical activity as time-dependent variables in the relative risk model (see e.g. Therneau and Grambsch 2000) adapted for random effects. This means that we modelled lifestyle factors with respect to occurrence of the first cancer and length of the follow-up period varied depending on participant’s age at first cancer. The impact of the main predictors (weight and physical activity) on cancer risk were assessed in separate models for types of cancer and separately for men and women. It is important to realize that age was neither a covariate nor a stratifying variable, but it was used as the outcome variable (time) together with a status variable indicating cancer/healthy as described in detail in the supplement. This allowed us to study for example if healthier style would associate with cancer occurrence at older age, as most of the LS carriers suffer from cancer at some age, but age at the first cancer occurrence varies greatly. We also performed additionally analyses to assessed whether there might be risk modification from the last recalled weight and physical activity measurement prior to diagnosis of cancer. For this analysis we dropped the data from the preceding measurements and used only the last timepoint measurements. In this analysis weight and physical activity are no longer time-dependent covariates because we only utilize measurements from one interval. Also, in this model we were able to account for the fact that subjects are of different age when cancer occurs. This occurs automatically in the counting process formulation, but at least three alternatives can be considered for the length-of-interval approach: 1) consider no impact from age, 2) use age for the start of interval as a predictor effect, and 3) stratify by age. For our current analysis, all three methods yield similar results.

To make the aim of our study clear, we modified the end of the introduction as follows:

Line 76-80: The aim of this retrospective study with longitudinal lifestyle and cancer register data was to investigate associations between body weight and physical activity on CRC and overall cancer risk among adult Finnish path_MMR men and women. We modelled the recalled levels of weight and physical activity as time-dependent variables in the relative risk model.

Ref. 45 in our manuscript: Therneau, T.M.; Grambsch, P.M. Modeling survival data: Extending the Cox model; 2000; ISBN 9781441931610.

Reviewer comment 2:

The systematic review have already reported that increased physical activity and lower body adiposity are also associated with decreased risk of cancers. Therefore, the findings in this study are not novel.

Author response:

We agree with the Reviewer that our findings are not completely novel. Higher physical activity and lower body adiposity have been shown to associate with lower cancer risk regarding different types of sporadic cancers. Our study investigated associations of adult life body weight and physical activity trajectories on cancer risk in LS population representing a population with genetic cancer predisposition syndrome, which currently have only limited amount of evidence is available. The systematic review conducted by Coletta et al. (2019) (ref 15) reported that increased physical activity and lower body adiposity decreases LS cancer risk. However, the reviewed evidence regarding physical activity comes from only two studies which were available at the time of preparing our manuscript. And we are not aware any other studies investigating physical activity and cancer risk in LS. We sincerely believe that within this context, more research is required among the distinct high cancer risk population of LS carriers. Hitherto, our study is the first to assess the associations of LS cancer risk and physical activity separately for men and women with LS.

Reviewer comment 3:

In Table 1, average ages were different among two group. Therefore, physical activity level can be different among them.

Author response:

We thank the Reviewer for this remark. We assume by ‘two groups’ the reviewer refers to healthy vs. cancer groups, and that the reviewer considers it to be a potential problem related to making inferences with respect to cancer risk ratios with respect to the physical activity categories. It is important to note that the Cox regression model compares group risks for the event (onset of cancer) at each timepoint (age in years in our analysis), not the groups of healthy vs. those diagnosed with cancer. Hence, while the imbalance in age means might be relevant in a logistic regression model comparing healthy/cancer (where age appears as a covariate), it is not relevant for the Cox regression analysis (where age forms the dependent variable together with the healthy/cancer status variable), where the importance is on the unfolding of physical activity categories over time (age). It is also important to keep in mind that this is an observational data analysis setting and, hence, physical activity participation may vary according to age. For these reasons, we chose to assess risk ratios using age as the time variable in the Cox regression model extended for time-dependent covariates and time-varying effects (more information can be found in Thereau & Grambsch 2000. Modeling survival data – extending the Cox model. Springer; see section 5.6 for time-dependent predictors and section 6.5 for time-varying effects). This model compares risks in PA categories in each age time-point and permits risk to vary from one time-point to the next. We found that we could simplify our analysis by dropping time-varying effects, as the risk remained sufficiently similar over time, which can also be seen from the percentages of subjects at risk in the PA-categories shown in the tables.

Men

Age

All cancers

LS

Inactive

Active

Inactive

Active

Healthy

[20, 30)

76

24

74

26

[30, 40)

80

20

80

20

[40, 50)

85

15

85

15

[50, 60)

89

11

89

11

[60, 70)

94

6

94

6

[70, End of follow-up]

100

0

-

-

Cancer

[20, 30)

75

25

78

22

[30, 40)

81

19

82

18

[40, 50)

87

13

88

12

[50, 60)

93

7

95

5

[60, 70)

100

0

100

0

[70, End of follow-up]

-

-

-

-

Women

Age

All cancers

LS

Inactive

Active

Inactive

Active

Healthy

[20, 30)

75

25

75

25

[30, 40)

72

28

71

29

[40, 50)

70

30

68

32

[50, 60)

81

19

77

23

[60, 70)

79

21

76

24

[70, EoFU]

80

20

82

18

Cancer

[20, 30)

75

24

78

22

[30, 40)

72

28

76

24

[40, 50)

65

35

68

32

[50, 60)

72

28

75

25

[60, 70)

78

22

89

11

[70, EoFU]

-

-

-

-

Reviewer comment 4:

The number of subjects with 60-77 years is too small to conclude.

Author response:

We appreciate the Reviewer’s comments. We understand that the reviewer may have understood that we split the analysis into age groups, but this was not the case. Age was the time variable for cancer incidence, and each subject could change their PA category or weight in ten-year leaps used as recall time points. We acknowledge the fact that most analysis in health / medical science tend to ignore time-dependency issues and kindly suggest the Reviewer to familiarize with time-dependent data analysis and construction of time-dependent variables in R using the brief online resource (available at: https://cran.r-project.org/web/packages/survival/vignettes/timedep.pdf). Now, for the Cox regression model, it is natural that the number of subjects at risk for cancer decreases over time. Thus, it is important to consider the ratio of risks and whether the risks differ over time; we did not find any substantive reason to conclude that such difference was important here for physical activity or weight predictors. It is also important to note that among 60-77 years there were 56 males and 70 females, which is by no means small for risk comparisons in the Cox regression model.

Reviewer comment 5:

It is not clear how healthy group was extracted.

Author response:

We apologize for being unclear how healthy group was extracted. All study participants carry pathogenic variants in mismatch repair genes known to predispose to high incidence of cancer. The healthy group included those LS participants who had not had any type of cancer during their life until data collection while the cancer group included those LS participants who had had previous cancer. Regarding participants with previous cancer, body weight and physical activity data was used only until the age of the first cancer occurrence. We have corrected this issue by adding the following:

Line 354-355: The healthy group included only LS carriers who had remained cancer free until data collection.

Reviewer comment 6:

The information of body weight, physical activity was collected by re-call. Therefore, recall bias should be considered.

Author response:

We are grateful for the feedback. All the information regarding body weight history and physical activity history was collected by re-call and thus there might be re-call bias. To assess the extent of recall bias would entail having measurements of weight and physical activity available, but if those variables had been available, we would have used them in our analysis. We have discussed this matter in detail in Discussion on lines 267-275. Encouraged by the Reviewer’s remarks, we have added the following:

Line 331-332: Taken together, we cannot exclude that re-call bias might have influenced the risk estimates.

Reviewer comment 7:

Overall, the findings in this study are not enough for the level of this journal.

Author response:

We appreciate the Reviewer’s view on our manuscript, but with respect, we disagree with this view. As we have already pointed out in our earlier responses, the earlier studies investigating association between long-term body weight and physical activity trajectories on cancer risk in LS population are rare. We are also first to report the associations of LS cancer risk and physical activity separately for men and women with LS. We sincerely believe that our findings are of interest for the readers and audience of the upcoming special issue “The Mechanisms of Physical Activity, Diet, and Body Composition in Cancer Prevention and Control” of Cancers journal. We also genuinely think that our manuscript meets the high standards of Cancers.

Reviewer 2 Report

I have the following comments:

Introduction

  • The introduction is insufficient, the second part should be improved. it should explain what background exists. The authors suggest that there is limited evidence to suggest that increased physical activity and decreased body adiposity is not correct. They should review the PREDIMED PLUS study. This study and the manuscripts derived from it would be a good example to look at.
  • They should explain prior to the aim of the study, what is known and why the need for the study.

Material and methods

  • The methodology is well described

Results

  • The results are well described

.Discusion

  • They have included a paragraph on limitations, but it would be appropriate to describe the strengths of this study (of which there are many).
  • Please check the format of the references included in the text.
  • There are many ideas from the discussion not referenced, please include them.
  • There are many comparisons with other studies, but one could add a reflection of the authors proposing that we are responsible for the reported results.

Conclusion

  • It is very scattered, please include the main ideas.

Author Response

Reviewer #2

I have the following comments:

  • Introduction

Reviewer comment:

The introduction is insufficient, the second part should be improved. it should explain what background exists. The authors suggest that there is limited evidence to suggest that increased physical activity and decreased body adiposity is not correct. They should review the PREDIMED PLUS study. This study and the manuscripts derived from it would be a good example to look at.

Author response:

We are thankful for the feedback and value the Reviewer’s remarks regarding these issues. We are aware that there are numerous reports and existing evidence concluding that increased physical activity and lower levels of body adiposity decrease cancer risk. Regarding this, we have carefully reviewed these reports (references 6-12, 15) in our manuscript. As the Reviewer suggested, we also reviewed the excellent PREDIMED-derived studies and found them interesting. However, in the context of Lynch syndrome (LS), we feel that the evidence is yet reasonably limited as most of the previously mentioned reports derive from the general public instead of path_MMR carriers. Due to the strong predisposition to cancers via inherited pathogenic variants in the mismatch repair genes, people with LS are not directly comparable to the general public or to sporadic cancer patients, when evaluating lifestyle factors and cancer risk. Moreover, as concluded in the Ref. 15 of our manuscript, the relationship between obesity or overweight and colorectal cancer risk among women with LS is controversial, and available evidence linking obesity and overweight to an increased endometrial cancer risk is insufficient. These matters have been investigated and discussed also, for example, in the following reports:

Endometrial cancer

  • Maenpaa J, Aaltonen M, Mecklin J, Staff S (2016) Evaluation of factors modifying endometrial cancer risk among women with Lynch syndrome: a cohort study. J Clin Oncol 34:e17113
  • Win A, Chau R, Dashti S et al (2014) Lifestyle modifiers of risk of endometrial cancer for women with germline mutations in DNA mismatch repair genes. Int J Gynecol Cancer 24(9 suppl 4):1577–1578. https://doi.org/10.1097/01.IGC.0000457075.08973 .89
  • Win A, Dowty J, Antill Y et al (2011) Body mass index in early adulthood and endometrial cancer risk for mismatch repair gene mutation carriers. Obstet Gynecol 117(4):899–905. https://doi. org/10.1097/AOG.0b013e3182110ea3

Colorectal cancer

  • Campbell P, Cotterchio M, Dicks E et al (2007) Excess body weight and colorectal cancer risk in Canada: associations in subgroups of clinically defined familial risk of cancer. Can- cer Epidemiol Biomark Prev 16(9):1735–1744. https://doi. org/10.1158/1055-9965.EPI-06-1059 (Ref. 22 in our manuscript)
  • Win A, Dowty J, English D et al (2011) Body mass index in early adulthood and colorectal cancer risk for carriers and non-carriers of germline mutations in DNA mismatch repair genes. Br J Cancer 105(1):162–169. https://doi.org/10.1038/bjc.2011.172

Furthermore, we are aware of only two studies (refs. 13,14) that have examined whether physical activity affects cancer risk estimates among individuals with LS. We sincerely think that more research is required regarding this matter, and therefore have stated in our manuscript that the evidence is limited.

In light of the reviewer comments, we have made the following changes:

Line 57-70: There is strong evidence derived from the general population that increased physical activity and reduced body adiposity are associated with decreased cancer risk (6-12). Recent research suggests that LS CRC risk can also be moderated by these lifestyle factors (13-15), but the number of studies that investigate the associations between lifestyle and LS cancer risk are scarce. Currently, there are only two studies that we are aware of that have assessed the association of physical activity and LS cancer risk (13,14) and have not taken into consideration that this potential association may vary during a lifespan. In addition, the evidence regarding the associations of cancer risk and obesity and overweight is contradictory among women with LS (15), and thus, the risk analyses should be performed separately for both sexes. It is of great importance to identify modifiable behavioral factors of LS cancer risk to motivate variant carriers to change suboptimal conduct or maintain healthy lifestyle. Lifestyle modification could efficiently aid in reducing individual cancer risk despite strong genetic predisposition.

Reviewer comment:

They should explain prior to the aim of the study, what is known and why the need for the study.

Author response:

We apologize that the background information preceding the aim of the study was insufficient. Encouraged by the Reviewer’s comments, we have made the following improvement (please also see our response to Reviewer #1, comment 1):

Line 76-80: The aim of this retrospective study with longitudinal lifestyle and cancer register data was to investigate associations between body weight and physical activity on CRC and overall cancer risk among adult Finnish path_MMR men and women. We modelled the recalled levels of weight and physical activity as time-dependent variables in the relative risk model.

  • Materials and methods

Reviewer comment:

The methodology is well described

Author response:

We thank the Reviewer for the encouraging words.

  • Results

Reviewer comment:

The results are well described

Author response:

We thank the Reviewer for the supportive comments. 

  • Discussion

Reviewer comment:

They have included a paragraph on limitations, but it would be appropriate to describe the strengths of this study (of which there are many).

Author response:

We thank the Reviewer for the supportive comments. We have discussed the strengths of this study in the Discussion section on Lines 316-322. However, encouraged by the Reviewer’s feedback, we have added the following:

Line 322-329: The observation period was initiated from the age of 20 years, instead of birth, to avoid detection of changes in body weight, which are merely due to natural growth and maturation. In doing so, we were also able to exclude the time-period when cancer incidence is extremely low even among the path_MMR carriers. We also chose to model the change in cancer risk in the near-term setting (during the age-period of cancer or censoring) as it could be more accurate than longitudinal change which could be influenced by poor re-call. We also used time-dependent covariate values which permit accounting for changes in predictor values over time.

Reviewer comment:

Please check the format of the references included in the text.

Author response:

We have checked the format of the references and corrected them according to the author instructions. The following corrections have been made:

Line 262: Kamiza et al. (2015)[13]

Line 266: physical activity and cancer risk on a near-term like Kamiza et al. (2015)[13],

Line 269: Further, the study by Dashti et al. (2018)[14]

Line 271: Unlike our study, Dashi et al. (2018)[14]

Line 277: Dashti et al. (2018)[14]

Reviewer comment:

There are many ideas from the discussion not referenced, please include them.

Author response:

We agree with the Reviewer that our use of references in the Discussion section was partly insufficient. Therefore, we have added the following references:

Line 292: Various reports in extant literature have suggested several mechanisms that link physical activity with reduced cancer risk[11].

  1. Friedenreich, C.M.; Neilson, H.K.; Lynch, B.M. State of the epidemiological evidence on physical activity and cancer prevention. Eur. J. Cancer 2010, 46, 2593–2604, doi:10.1016/j.ejca.2010.07.028.

Line 293: For example, physical activity produces multiple beneficial changes in cardiorespiratory systems[33],

  1. Lavie, C.J.; Ozemek, C.; Carbone, S.; Katzmarzyk, P.T.; Blair, S.N. Sedentary Behavior, Exercise, and Cardiovascular Health. Circ. Res. 2019, 124, 799–815, doi:10.1161/CIRCRESAHA.118.312669.

Line 294: and being physically active also helps with weight control as well as to reduce excess adiposity[12].

  1. Moore, S.C.; Lee, I.M.; Weiderpass, E.; Campbell, P.T.; Sampson, J.N.; Kitahara, C.M.; Keadle, S.K.; Arem, H.; De Gonzalez, A.B.; Hartge, P.; et al. Association of leisure-time physical activity with risk of 26 types of cancer in 1.44 million adults. JAMA Intern. Med. 2016, 176, 816–825, doi:10.1001/jamainternmed.2016.1548.

Reviewer comment:

There are many comparisons with other studies, but one could add a reflection of the authors proposing that we are responsible for the reported results.

Author response:

We thank the Reviewer for appreciating our thorough discussion section comparing our results with the current state-of-the-art on the field. We have done our best to use terms such as ‘in our study’, in the current study’, ‘we found’ and ‘our findings’ to help readers to differentiate our own results from the existing previous studies. To make this approach even more consistent, the following changes were made to the manuscript:

Line 275: Near-term cancer risk assessment, which was used in the current study, may be particularly effective for identifying specific suboptimal lifestyle changes that could be associate with carcinogenesis and precede cancer occurrence.

Line 316: A major strength of the current study is that the study cohort comprised participants who have undergone comprehensive screenings of LS-predisposing mutations, ascertainment utilizing Amsterdam and Bethesda clinical criteria and cascade testing, and those who have been offered colonoscopy surveillance at 2–3-year intervals.

Conclusion

Reviewer comment:

It is very scattered, please include the main ideas.

Author response:

We thank the Reviewer for highlighting an important issue. We have made the following changes to Conclusions:

  • We have moved the part

 “In addition to regular cancer screenings, it is important to follow potential lifestyle-related modifiable risk factors—such as body weight, physical activity, and nutrition—among path_MMR carriers during their regular health care visits. Optimal lifestyle could partially compensate the strong genetic predisposition to cancers. Nowadays, individual cancer risk can be calculated and demonstrated via online tools (www.plsd.eu), which can be used to improve motivational support for healthy lifestyle maintenance or lifestyle changes.”

of the original Conclusions to lines 302-316 with following changes:

Line 302-316: As described previously in our study, the associations of decreased cancer risk and healthy lifestyle in the general population has been observed in large population-based studies. Since the influence of healthy behaviour on decreased cancer risk was observed in our study with a limited number of participants, it could be possible, that the effect of the modifiable behavioural risk factors – physical activity and body weight – is emphasised in path_MMR carriers due to their strong genetic predisposition to cancers. Therefore, it is important to follow these modifiable risk factors among path_MMR carriers during their regular health care visits. Optimal lifestyle could partially compensate the strong genetic predisposition to cancers and thus help in cancer prevention. Nowadays, individual cancer risk can be calculated and demonstrated via online tools (www.plsd.eu), which can be used to improve motivational support for healthy lifestyle maintenance or lifestyle changes.

and

 2) added the parts

 “Taken together, our results emphasize the importance of weight maintenance and high-intensity physical activity throughout the lifespan in cancer prevention in men with path_MMR.”

and

 ”The results of our study could be used in developing the cancer risk quantification methodology based on the consideration of various risk factors that are modifiable by the individuals themselves.”

to the end of the current Conclusions section. Thus, the current Conclusions section now appears as following:

Line 468-479: To conclude, our results suggest that particularly men with path_MMR were susceptible to lifestyle exposures that may be either protective or hazard increasing. We found that weight gain over adulthood increased the risk of cancer in men, while participating in more intense physical activity across lifespan may have a cancer-preventive effect. According to our results, women were not as prone to lifestyle-related risk factors. The sex-based difference in the associations could be explained by differences in weight gain, which was smaller in women, and by sex-related factors modifying body composition over time. Taken together, our results emphasize the importance of weight maintenance and high-intensity physical activity throughout the lifespan in cancer prevention in men with path_MMR. The results of our study could be used in developing the cancer risk quantification methodology based on the consideration of various risk factors that are modifiable by the individuals themselves.

Reviewer 3 Report

This manuscript is another example of the impressive Finnish database on Lynch syndrome, showing the link between weight, physical activity and carcinoma risk in as much detail as current data allow.

In my view, there is only one point of criticism left after the latest corrections:

Page 14, lines 210-213 you write: “Intriguingly, we also found that near-term weight increase had a CRC-protective effect in women. To speculate, it is possible that an undiagnosed cancer during the near-term period could have resulted in weight decrease among some of those who were diagnosed with cancer at the beginning of the next time interval; ….”

Weight change under these circumstances affects women and men equally. In this respect, it is not speculative but rather non sequitur.

I would either delete the paragraph or the speculation should really be gender-related.

One speculative possibility would be that gynoid fat distribution might be even cancer-protective.

As you wrote, premenopausal women do not have visceral fat so much as gynoid fat distribution. If this gynoid fat distribution were even carcinoma-protective under certain circumstances, it would make sense if weight gain had carcinoma-protective consequences.

On the contrary, as you also wrote visceral fat (which is prevalent in men throughout their lives) is associated with an increased risk of cancer and therefore increased physical activity in men throughout their lives reduces the risk of carcinoma.

Minor issue:

It is probably a small grammatical error:

Our results suggest that associations between lifestyle and cancer risk are differ between men and women and may vary during the course of life.

Author Response

Reviewer #3

This manuscript is another example of the impressive Finnish database on Lynch syndrome, showing the link between weight, physical activity and carcinoma risk in as much detail as current data allow. In my view, there is only one point of criticism left after the latest corrections:

Page 14, lines 210-213 you write: “Intriguingly, we also found that near-term weight increase had a CRC-protective effect in women. To speculate, it is possible that an undiagnosed cancer during the near-term period could have resulted in weight decrease among some of those who were diagnosed with cancer at the beginning of the next time interval; ….”

Weight change under these circumstances affects women and men equally. In this respect, it is not speculative but rather non sequitur. I would either delete the paragraph or the speculation should really be gender-related. One speculative possibility would be that gynoid fat distribution might be even cancer-protective.

As you wrote, premenopausal women do not have visceral fat so much as gynoid fat distribution. If this gynoid fat distribution were even carcinoma-protective under certain circumstances, it would make sense if weight gain had carcinoma-protective consequences.

On the contrary, as you also wrote visceral fat (which is prevalent in men throughout their lives) is associated with an increased risk of cancer and therefore increased physical activity in men throughout their lives reduces the risk of carcinoma.

Author response:

We are thankful for the feedback. As the Reviewer points out, weight change due to cancer should be similar for men and women. Thus, we agree with the Reviewer that possibility of the hidden cancer to affect our results only in one sex is not very likely scenario. A great number of evidence exists stating that there are major differences in pre- and post-menopausal cancer risk. There is also conclusive evidence indicating that visceral fat is more strongly associated with increased colorectal cancer risk than peripheral fat in both genders. Therefore, we find the Reviewers views and suggestions very interesting. In our study a great majority of the female participants’ cancer events occurred after the age of 60 years. Thus, there is a possibility that most of the diagnosed female participants were already post-menopausal and therefore the gained weight could have been more of an android-type rather than gynoid-type. If so, the suggestion of gynoid-type of fat distribution as a protective element for cancer during the age-period of colorectal cancer diagnosis is not consistently supported by our data although we cannot exclude the possibility that potential premenopausal lower amount of visceral fat (from early adulthood until 50’s) compared to men can have long-term protective effects. When considering the individual differences in menopausal factors such as age and fat distribution, and individual differences in body composition, we agree with the Reviewer that it is possible that gynoid fat distribution might have affected the risk estimates. On the other hand, estrogen-containing hormone therapy has been indicated to be protective, neutral or to increase a risk for colorectal and cervical cancer while it seems to increase the risk of breast cancer. To conclude, the influence of post-menopausal exogenous estrogen is likely to be cancer-type specific, which makes it difficult to speculate the role of hormonal differences in our study where we did have opportunity to measure hormones and have not investigated the use of hormone therapy. Nevertheless, adipose tissue is known to synthesize estrogen through androgen conversion (Simpson et al 1996), thus, theoretically, higher adiposity could provide some protection through higher estrogen levels although it is unknown if adipose-derived estrogen has true systemic effects. Therefore, there lies a possibility that the plausible protective effects of estrogen derived from adipose tissue might have masked the effect of weight gain on colorectal cancer risk.

We have made the following changes:

Line 217-230: Intriguingly, we also found that near-term weight increase had a CRC-protective effect in women. To speculate, it is possible that hormonal factors might have influenced the risk estimates. The primary source of endogenous estrogen in post-menopausal women has been suggested to be adipose tissue [32,33]. Therefore, higher adiposity could maintain higher systemic estrogen level, which in turn may provide some protection through for instance anti-inflammatory actions of estrogens [34]. However, this is highly speculative. Evidence on exogenous estrogen use indicates estrogen use to be cancer protective, neutral or increasing the risk of different cancers [35-37]. In the current study, we did not investigate potential role of hormone therapy nor had the possibility to measure estrogen levels, thus we cannot exclude the role of systemic estrogen level. Therefore, as the great majority of CRC diagnosed women in our study were over 60 years of age, there lies a possibility that the plausible protective effects of estrogen derived from adipose tissue might have masked the effect of weight gain on colorectal cancer risk.

References:

  1. Simpson ER, Bulun SE, Nichols JE, Zhao Y. Estrogen biosynthesis in adipose tissue: regulation by paracrine and autocrine mechanisms. J Endocrinol 1996;150 Suppl:S51 – 7.
  2. Nelson, L.R.; Bulun, S.E. Estrogen production and action. Am. Acad. Dermatol. 2001, 45, S116-24, doi:10.1067/mjd.2001.117432.
  3. Straub RH. The complex role of estrogens in inflammation. Endocr Rev. 2007 Aug;28(5):521-74. doi: 10.1210/er.2007-0001. Epub 2007 Jul 19. PMID: 17640948.
  4. Campbell, P.T.; Newcomb, P.; Gallinger, S.; Cotterchio, M.; McLaughlin, J.R. Exogenous hormones and colorectal cancer risk in Canada: associations stratified by clinically defined familial risk of cancer. Cancer Causes Control 2007, 18, 723–733, doi:10.1007/s10552-007-9015-7.
  5. La Vecchia, C.; Gallus, S.; Fernandez, E. Hormone replacement therapy and colorectal cancer: an update. Br. Menopause Soc. 2005, 11, 166–172, doi:10.1258/136218005775544264.
  6. Liang, J.; Shang, Y. Estrogen and cancer. Rev. Physiol. 2013, 75, 225–240, doi:10.1146/annurev-physiol-030212-183708.

Minor issue: It is probably a small grammatical error:

Our results suggest that associations between lifestyle and cancer risk are differ between men and women and may vary during the course of life.

Author response:

We are thankful for pointing out this small grammatical error.

We have corrected this issue:

Line 174-176: Our results suggest that associations between lifestyle and cancer risk differ between men and women and may vary during the course of life

Round 2

Reviewer 1 Report

This manuscript has been revised. The authors stated several limitations according to my previous comments. However, these modifications are not enough for the level of this journal.

Major points

  1. The systematic review have already reported that increased physical activity and lower body adiposity are also associated with decreased risk of cancers. Therefore, the findings in this study are not novel.
  2. In Table 1, average ages were different among two group. Therefore, physical activity level can be different among them.
  3. The number of subjects with 60-77 years is too small to conclude.

4. The information of body weight, physical activity was collected by re-call. Therefore, recall bias should be considered. Although the authors stated that they are limitation, the main outcome was based on these information, therefore, this limitation is critical.

Author Response

Dear Reviewer,

With respect, we yet disagree with the Reviewers view that our modifications are not enough for the level of Cancers. We strongly believe, that we have answered comprehensively to the major points highlighted by the Reviewer. Therefore, we have nothing more to add to our previous comments.

Reviewer 2 Report

The authors have responded very well to the suggestions. I have nothing more to add

Author Response

Dear Reviewer,

We are thankful for the feedback. We also have nothing more to add to our previous comments.